# Anal human papillomavirus infection among men who have sex with men and transgender women living with and without HIV in Pakistan: findings from a cross-sectional study

Muslima Ejaz ,[1,2] Soren Andersson,[3] Salma Batool,[4] Tazeen Ali,[5] Anna Mia Ekström[1]

This work has been presented in 32nd International Papillomavirus Conference in conjunction with AOGIN 2018, being held in Sydney Australia from 2 October 2018 to 6 October 2018. This conference attendance was part of my Doctoral Programme and financially being supported by Faculty Development Award by The Aga Khan Hospital & Medical University Pakistan.

For numbered affiliations see end of article.

**Correspondence to**
Dr Muslima Ejaz;
muslima.ejaz@aku.edu

## ABSTRACT

**Objectives** The aim of this study was to determine the prevalence of infection, genotypes and risk factors for human papillomavirus (HPV) among men who have sex with men (MSM) and transgender women living with and without HIV in Pakistan. Anal infection with HPV is very common worldwide among MSM, particularly among MSM living with HIV. The high prevalence of HIV among MSM and male-to-female transgendered individuals in Pakistan is a significant health concern since access to screening and health-seeking is often delayed in this stigmatised key population.

**Design** This cross-sectional study was conducted between March 2016 and November 2017.

**Participants, setting and data collection** This study recruited MSM and transgender-women who self-reported to have had anal sex in the last 6 months, and were at least 18 years of age, from the sexual health and antiretroviral therapy centres. Structured questionnaires were administered, and blood samples were obtained to confirm HIV status. Anal swabs were collected for HPV-DNA detection and typing.

**Main outcome measures** The primary outcome was the prevalence of 'HPV-DNA infection'. The prevalence ratios (PR) were calculated using Cox proportional hazard model algorithms to analyse the association between exposure variables and HPV-infection.

**Results** Complete data were available for 298 MSM and transgender women (HIV +n=131; HIV−n=167). The overall HPV-DNA prevalence was 65.1% and was higher in participants living with HIV as compared with HIV-negative (87% vs 48%; $\chi^2$p≤0.001). Likewise, 28.9% of participants living with HIV were infected with two or more than two types of HPV as compared with 18.8% participants without HIV($\chi^2$ p≤0.001). The most frequent HPV type was HPV6/11 (46.9%), followed by HPV16 (35.1%), HPV18 (23.2%) and HPV35 (21.1%). HIV status (PR 2.81, 95% CI 2.16 to 3.82) and never condom use (PR 3.08, 95% CI 1.69 to 5.60)) were independently associated with prevalence of 'anal-HPV16 infection' when adjusting for confounding for age, other sexual and behavioural factors, for example, smoking and alcohol consumption.

## Strengths and limitations of this study

► This study is the first description of the epidemiology of anal human papillomavirus (HPV) infection among sexual and gender minorities (men who have sex with men, and transgender women) living with HIV in Pakistan.

► Given the strong stigma, homophobia and discrimination against sexual minorities in Pakistan, data collection included a careful process of gaining trust among study participants.

► The results may be of a public health importance in Pakistan, indicating the need for gender-neutral HPV vaccination before sexual debut, targeting both women and men.

► The cross-sectional study design precludes any conclusion on temporal relationships between potential risk factors and the outcome.

► Any under-reporting or over-reporting regarding sexual behaviours or lifestyle factors leading to information bias cannot be excluded.

**Conclusion** High prevalence of HPV indicates a substantial future risk of anal cancer in Pakistani MSM and transgender women, and particularly in those living with HIV. Current findings support anal Pap-smear HPV screening for this particular group and vaccination efforts for future generations.

## INTRODUCTION

Human papillomaviruses (HPV) is the most common sexually transmitted infection (STI) worldwide,[1] and is rapidly acquired after sexual debut.[2] A meta-analysis of global data on HPV in men who have sex with men (MSM) found prevalence rates of anal HPV to be 93% in MSM living with HIV and 64% in HIV-negative MSM, and these rates did not decrease with increasing age.[3] More than 90 genotypes of HPV have been identified and classified either as low-risk (LR) types

(predominantly HPV6 and HPV11) causing genital warts. High-risk (HR) HPV types (predominantly HPV16 and HPV18) cause cancer of the cervix,[4] its pathogenic role has also been demonstrated by others with oropharyngeal cancers.[5 6] Persistent infection with HPV16 is the major cause of anal cancer.[7 8] While the burden of HPV-related disease in all men is lower than in all women, certain at-risk groups such as MSM and transgender women,[9 10] and particularly those with HIV infection,[7] are at HR for HPV infection, and related anal cancer and its precursor high-grade squamous intraepithelial lesions.[11] The reported incidence of anal cancer is 1–2/100 000 in the general population,[12] while in MSM it has been reported up to 35 cases per 100 000[13] and even up to 131/100 000 in MSM living with HIV[14 15] regardless if they are on antiretroviral treatment or not.[16–18]

In Pakistan, the demography of MSM is broadly classified as: MSM who are in relationships but outwardly closeted; as male sex workers (MSW) who are males and undertakes receptive anal or oral sexual activity with a man in return of money or other financial benefits; and as hijras (transvestites/transgender women) who are biologically male with male genitals but function in society as if female and take the receptive role in anal sex.[19 20] The 'hijra' is a South-Asian identity traced back many centuries to India, where they were held in high-esteem and were political advisors to kings and bodyguards to queens. Today, however, they often face social exclusion and many engage in commercial sex work.[21]

Pakistan is an authoritarian Islamic republic with conservative social values, particularly on issues of sex. MSM, and in particular MSW, face high levels of discrimination and contempt from society and are subject to harassment, sexual violence and detention by the state authorities. Their health-seeking behaviours are often poor due to stigmatisation in healthcare settings and for the fear of being identified. As a result, they are reluctant to seek healthcare not only in public setups but also in private facilities. One previous study on sex work and MSW in Pakistan showed that the majority of MSWs are poor, have little knowledge about safe sex, and cannot negotiate the use of condoms and safe sex with their paying clients, predisposing them to acquiring STIs.[20]

Pakistan's national government has lately recognised MSM as a vulnerable population, and prevention and control programmes targeting MSMs in its big cities are being organised (of the estimated 46 264 MSM in Pakistan, Karachi is home to an estimated 18 361 MSM, according a survey conducted for mapping key populations in 2015–2016).[22] However, programme service delivery, such as HIV counselling and testing and other STI screening services, is performed by non-governmental and community-based organisations. They are dependent on national and international funders, which has led to fragmented and incomplete coverage of preventive services for this key population.[23 24] Furthermore, the prevalence of HIV in these groups is reaching alarming proportions. Results from the fifth round of Integrated Behaviour and Biological second-generation surveillance for HIV/AIDS in Pakistan 2016–2017 reported over 9% HIV prevalence in MSM and almost 13% in transgender women living in Karachi.[22]

So far there is no research on the burden of HPV infection and HPV-related anal squamous intraepithelial lesions in this key population in Pakistan. Currently, HPV vaccines, in which the bivalent vaccine protects against HPV16/18 and quadrivalent vaccine protects against HPV16/18 and the LR types HPV6/11,[25] are available in pharmacies in Pakistan but are not yet made available to the target population. The Ministry of Health has not yet included it in their national immunisation programme for any group in society.

We conducted a cross sectional study in Karachi Pakistan in 2016–2017 with the objective to determine the prevalence and risk factors for anal HPV infection including sexual risk behaviours and the association with HPV infection in this dynamic population to inform future policy and prevention strategies.

## METHODS

### Setting and participants

Enrolment occurred in two phases for this cross-sectional study. The initial recruitment (from March 2016 to May 2016) started at the ART clinic with transgender women, a centre run by the National AIDS Control Programme of Pakistan in Civil Hospital Karachi (CHK)—the largest (1900 bed) tertiary care public hospital of Karachi Pakistan (recruited n=33). Low pace, logistic issues and privacy concerns led to the relocation and setting-up of recruitment centre at a sexual health clinic run by a community-based organisation, Perwaaz Trust Karachi Pakistan, where the study population was reached using snowball technique and peer referrals. During the second phase between April 2017 and November 2017 a total of 265 study participants with different sexual orientations, that is, MSM (homosexuals, bisexuals) and transgender women living with and without HIV were recruited. Participants were eligible for the study who self-reported to have had anal sex in the last preceding at least 6 months and were age 18 and above.

### Patient and public involvement

No patients or member of the public were involved in the design, conduct, reporting or dissemination plans of our study. Data were kept confidential and anonymous and were used for research purposes only.

### Study procedure

Verbal and written informed consent were obtained from all eligible and willing participants for all study procedures. For participants who could not read or write their name, a thumb impression was obtained. Participants were informed that they were free to interrupt the interview or withdraw from the study at any time without any negative consequences in terms of clinical services provided

to them. A 35–40 min long structured questionnaire was administered at a private space to interview the study participants in order to collect data concerning sociodemographic information, sexual and reproductive history and medical history relevant to HIV, any anal disease and use of ART. A blood sample was obtained for confirmation of their HIV status (through Architect HIV Ag-Ab Combo kit by ABBOTT), viral load (real time PCR Artus Qiagen) and CD4 +T cell count (through FACS Count TM flow cytometry, Becton Dickinson, Franklin Lakes, New Jersey, USA). Two anal swabs for HPV testing were collected by a trained general physician. A water-moistened swab was inserted 5–6 cm proximal to the anal verge and the samples were obtained from anorectal transition zone around the dentate line in the anal canal. The swab was then agitated vigorously in a tube containing 3 mL of methanol-based fixative—a Sample Transport Medium (UTM-RT viral transport media with flocked polyester swabs, Copan Diagnostics, Corona, California, USA) and transported to the lab. In Molecular lab the solution was stored at −70°C before further processing and was later used for the extraction and detection of HPV DNA by the PCR. The DNA from rectal swab was extracted by using Qiagen DNA mini kit and DNA was eluted in 80 uL of elution buffer and stored at −80°C for further use. HPV DNA was detected by using MY09/MY11 primers from L1 region resulted in amplification of 450 bp of L1 gene. PCR for Housekeeping gene glyceraldehyde 3-phosphate dehydrogenase (GAPDH) was done for every extracted DNA and samples tested negative for GAPDH gene were excluded from study. Nested multiplex PCR assay was used[26] for detection of HPV subtypes included LR types 6/11 and HR (group 1) types 16, 18, 31, 33, 35, 52, 56, 58 and 59 as described by Sotlar *et al*.[27] Primers for first-round PCR for GP-E6/E7 consensus sequence and second round PCR for HPV subtypes 16, 18, 6/11 31, 33, 35, 52, 56, 58 and 59 were synthesised from Eurofins MWG/Operon Germany. The HPV subtype specific primers were used in four cocktails and size of the nested PCR product was used for the identification of each HPV type by gel electrophoresis.[28]

### Sample size calculation

For the estimation of HPV prevalence, a sample size of 267 participants were needed with an anticipated prevalence of HPV of 50%, 6% precision and 5% level of significance. For the analysis of factors associated with HPV infection, we aimed for a sample of at least 256 participants to achieve 80% power with a range of exposures between 9% and 39% and an anticipated prevalence ratio (PR) of 2.5 or more with 5% level of significance. An additional sample of 10% was added to cater for inadequate biological samples.

### Data management and statistical analysis

The database was analysed using Statistical Package for Social Sciences (IBM SPSS Statistics for Windows, V.24.0, IBM).

The mean value was calculated with standard deviation (SD) for symmetrically distributed variables and independent sample t-test was used for statistical significance, while the median with IQRs was used for asymmetrical quantitative variables. For categorical variables, proportions were employed; for comparison of proportions between groups, $\chi^2$ or Fisher's exact test (where needed) were used for p values. The overall HPV DNA prevalence (as determined by PCR) is reported as the proportion of HPV positive participants among all included individuals. Conversely, type-specific HPV positivity is presented as the proportion of HPV-subtype positive individuals divided by all HPV positive individuals.[7] The CIs for the proportions were estimated using binomial distribution (n,p) assuming a Gaussian error. Prevalence of 'any anal HPV infection' and 'any anal HR-HPV infection' stratified by HIV status for all reported sexual orientations was also assessed using Pearson $\chi^2$. For multiple comparisons for each sexual orientation stratified by HIV status for 'any anal HPV infection' and 'any anal HR-HPV infection' Bonferroni correction method was used, the p value threshold was $(\alpha/n)=0.05/6=0.0083$.

Since the design of the study was cross-sectional and the outcome of the interest was common (HPV prevalence); for the association between independent risk factors and the outcome variables 'any anal HPV infection' that is, all those HPV types that we identified in our study sample and 'anal HPV16 infection', we employed a Cox proportion hazard model algorithm to estimate crude and adjusted PRs and their 95% CIs in univariate and multivariate models'. Moreover, we employed robust SEs to get the closest CIs. A significance threshold for the p<0.05 was used for the final model.

## RESULTS

### Study population characteristics

The final analysis included 298 MSM and transgender women (235 MSM and 63 transgender women) to assess the prevalence of HPV DNA in the anal canal. Out of these 298, 131 (44%) had an HIV-infection and 167 (56%) were HIV-negative. Among MSM living with HIV, 33 (25%) were transgender women. Almost 90% of the participants living with HIV were on ART and two thirds of them had an undetectable HIV viral load.

The overall mean age (±SD) of the 298 recruited participants was 28.8 (±8.06). The level of education was low: 28% reported no formal education and 12% less than primary level. Most respondents were currently not married. Among study participants almost 78% were smokers with mean age of smoking initiation 16.5 (±4.9) and duration of 12 (±8.5) years. Almost 80% were sex workers, with 12.4 (±8.5) years mean duration of sex work. The mean age at sexual debut was 16.4 (±4.7) years. Only 24.2% reported consistent condom use during sex (table 1).

MSM with any anal HPV infection (n=194) were younger than MSM without detectable HPV infection (28.9 (±7.3)

**Table 1** Sociodemographic and lifestyle characteristics of MSM and transgender women in Karachi, Pakistan by HPV status

| Characteristics | Overall (n=298) | HPV positive (n=194) | HPV negative (n=104) | P value* |
|---|---|---|---|---|
| Age categories in years n (%) | | | | 0.254 |
| <25 | 102 (34.2) | 72 (37.1) | 30 (28.8) | |
| 25–29 | 90 (30.2) | 59 (30.4) | 31 (29.8) | |
| 30–34 | 41 (13.8) | 27 (13.9) | 14 (13.5) | |
| >35 | 65 (21.8) | 36 (18.6) | 29 (27.9) | |
| Education n (%) | | | | 0.334 |
| None | 83 (27.9) | 56 (29.2) | 27 (26.2) | |
| Primary school | 36 (12.1) | 29 (15.1) | 07 (6.8) | |
| Middle school | 41 (13.8) | 24 (12.5) | 17 (16.5) | |
| Matric school | 78 (26.2) | 49 (25.5) | 29 (28.2) | |
| Intermediate | 38 (12.8) | 23 (12.0) | 15 (14.6) | |
| Graduate | 19 (6.4) | 11 (5.7) | 08 (7.8) | |
| Marital status n (%) | | | | 0.256 |
| Unmarried | 234 (78.5) | 156 (80.4) | 78 (75.0) | |
| Married | 57 (19.1) | 32 (16.5) | 25 (24.0) | |
| Separated/divorced | 7 (2.3) | 06 (3.1) | 01 (1.0) | |
| Smoking status n (%) | | | | 0.009 |
| Yes | 232 (77.9) | 160 (82.5) | 72 (69.2) | |
| Average no of cigarettes/day n (%) | | | | 0.035 |
| <10 | 128 (55.2) | 82 (52.9) | 46 (63.9) | |
| 10–20 | 79 (34.1) | 52 (33.5) | 24 (33.3) | |
| >20 | 25 (10.7) | 21 (13.5) | 2 (2.8) | |
| Alcohol drinking: n (%) | | | | 0.555 |
| Yes | 71 (23.8) | 44 (23.4) | 37 (20.4) | |
| Sexual behaviour: n (%) | | | | 0.127 |
| Bisexual | 68 (22.8) | 39 (20.1) | 29 (27.9) | |
| Homosexual | 167 (56.0) | 108 (54.6) | 59 (56.71) | |
| Transgender | 63 (21.2) | 47 (25.3) | 16 (15.4) | |
| Male sex workers: n (%) | 237 (79.5) | 155 (79.5) | 82 (78.8) | 0.83 |
| Anal receptive sexual partners during the last 6 months† n (%) | | | | 0.004 |
| <25 | 25 (8.4) | 16 (8.7) | 9 (10.2) | |
| 25–50 | 36 (12.1) | 17 (9.3) | 19 (21.6) | |
| 51–100 | 89 (29.9) | 56 (30.6) | 33 (37.5) | |
| >100 | 121 (40.6) | 94 (51.4) | 27 (30.7) | |
| Condom use during anal receptive sex n (%) | | | | <0.001 |
| Consistent use | 72 (24.2) | 15 (7.7) | 49 (47.1) | |
| Inconsistent use | 175 (58.7) | 141 (72.7) | 41 (39.4) | |
| Never | 51 (17.1) | 38 (19.6) | 14 (13.5) | |
| Preferred anal sex role n (%) | | | | 0.009 |
| Mainly receptive | 264 (88.6) | 173 (89.2) | 81 (77.9) | |
| Mainly insertive | 34 (11.4) | 21 (10.8) | 23 (22.1) | |
| Other STI (gonorrhoea, Trichomonas) n (%) | 167 (56.0) | 133 (68.6) | 34 (32.7) | <0.001 |
| Recruitment settings n (%) | | | | 0.566 |
| ART | 33 (11) | 20 (10.3) | 13 (12.5) | |
| CBO | 265(89) | 174 (89.7) | 91 (87.5) | |

Continued

**Table 1** Continued

| Characteristics | Overall (n=298) | HPV positive (n=194) | HPV negative (n=104) | P value* |
|---|---|---|---|---|

*χ2 test was used for p values.
†Missing values=9.1%.
ART, Anti retroviral therapy; CBO, Community based organisation; HPV, human papillomavirus; MSM, men who have sex with men; STI, sexually transmitted infection.

vs 30.2 (±9.1) years; independent sample t-test p=0.027). Smoking was more prevalent among HPV-positive MSM (82.5% vs 69.2%) than among those uninfected, with a significant difference in mean age (±SD) of 15.6 (±4.5) years at smoking initiation. The mean age (±SD) at sexual debut among study participants with an HPV infection was 15.7 (±4.3) vs 17.6 (±5.2) (independent sample t-test-p=0.001). HPV-infected had more sexual partners, more inconsistent condom users and more often practised receptive anal sex (table 1).

MSM living with HIV were, however, older than MSM who were HIV negative (mean (±SD): 30.9 (±0.70) vs 27.2 (±0.59) independent sample t-test p≤0.001), more frequently smokers ($\chi^2$p=0.01), younger at initiation of smoking (independent sample t-test p≤0.001) and had smoked for a longer duration of time (($\chi^2$p≤0.001). They also reported younger age at sexual debut (independent sample t-test p≤0.001), had more sexual clients ($\chi^2$p≤0.001), were mainly the receptive anal partner ($\chi^2$p≤0.001) and used condoms less consistently than their HIV negative counterparts ($\chi^2$p≤0.001).

### Sexual orientation of the study sample
Among participants living with HIV, 45.1% reported their sexual orientation as homosexual, 29.8% as bisexuals and 25.1% as transgender women. The sexual orientations reported by HIV uninfected participants were, 64.7% as homosexuals, 17.3% as bisexuals and 18.0% as transgender women.

### HPV DNA prevalence and genotype distribution
Overall, 65.1% (n=194/298) of the total study population was positive for HPV DNA (95% CI 59.5% to 70.3%). MSM living with HIV had a significantly higher prevalence of HPV (87% (95% CI 80.0% to 91.0%)) compared with those who did not have HIV (48% (95% CI 40.0% to 55.0%)) as well as any oncogenic anal HPV infection (83.2% (95% CI 75.8% to 88.6%) verses (35.3% (95% CI 28.4% to 42.8%). Likewise, 28.9% of participants living with HIV were infected with two or more than two types of HPV as compared with 18.8% participants without HIV ($\chi^2$p≤0.001). The most common HPV types were HPV6/11 (46.9%), HPV16 (35.1%), HPV18 (23.2%), HPV35 (21.1%) and HPV58 (8.2%). Moreover, the distribution of HPV6/11 was more among HIV uninfected participants (61.3%) as compared with HIV infected participants (36.8%) (($\chi^2$p=0.001). However, HPV16 was more common among HIV infected participants (44.7% verses 21.2%) and the difference was statistically significant ($\chi^2$p=0.001) (table 2).

### HPV DNA prevalence by HIV status and sexual orientation
For participants living with HIV, the prevalence of 'any anal HPV infection' were 88.1% for homosexual men, 76.9% for bisexual men and 96.9% for transgender women and the prevalence for 'Any anal HR-HPV infection' was 74.5%, 51.3% and 81.8%, respectively ($\chi^2$p=0.001) (figure 1).

For HIV uninfected participants, the prevalence of 'any anal HPV infection' was 50.9% for homosexual men, 31.0% for bisexual men and 53.4% for transgender women and the prevalence for 'any anal HR- HPV infection' was 37.1%, 20.7% and 43.3%, respectively. ($\chi^2$p=0.001).

Post hoc analysis with the Bonferroni correction method determined that the homosexuals were significantly different with being infected with 'any anal HPV infection' and 'any anal HR-HPV infection' ($\chi^2$p=0.0014 and 0.0019) when compared with bisexuals and transgender women living with and without HIV.

### Independent variables associated with anal HPV infection
Being HIV positive (PR 2.81 (95% CI 2.16 to 3.82)) and inconsistent/never condom use (PR 2.61 (95% CI 1.50 to 4.50)/(PR 3.08 (95% CI 1.69 to 5.60) were independent risk factors associated with increased prevalence of HPV16 infection, adjusted for all other demographic, sexual and behavioural risk factors (table 3).

### DISCUSSION
To the best of our knowledge, this is the first report on anal HPV infection among MSM living with and without HIV in Pakistan. The high prevalence of anal HPV among MSM living with HIV (87%) in Pakistan is similar to Thailand (85.0%)[29] and China (81%–82%),[30 31] slightly lower than in neighbouring India (95%)[32] and Canada (98%),[33] but significantly higher than what has been reported from the Netherlands (65%).[34] In HIV-negative MSM, however, studies previously carried out in the USA and Latin America showed similar prevalence levels of HPV prevalence ranging from 42% to 66%, which is comparable to our 48%.[35–37] In Pakistan, there is heterogeneity among MSMs, and it is predominantly MSWs who register with the community-based organisations with a hope to get free medical services of any kind in a secluded setting. It is MSW who accounted for 80% of our study population, partly explaining the differences in the prevalence of anal HPV infection between our study and others. In a study from India, which has similar social and cultural values, Hernandez et al[32] noticed HPV prevalence of 98%

**Table 2** Distribution of HPV types among MSM and transgender women in Karachi, Pakistan stratified by HIV status

| HPV types | ALL n (%) | HPV among HIV positive n (%) | HPV among HIV negative n (%) | P value* |
|---|---|---|---|---|
| Any type | 194 (65.1) | 114 (58.8) 87.0 | 80 (41.2) 48.0 | <0.0001 |
| HR types | | | | |
| HPV 16 | 68 (35.1) | 51 (44.7) | 17 (21.2) | 0.001 |
| HPV 18 | 45 (23.2) | 30 (26.3) | 15 (18.8) | 0.219 |
| HPV 31 | 14 (7.2) | 8 (7.5) | 6 (7.0) | 0.898 |
| HPV 33 | 13 (6.7) | 13 (11.4) | 0 (0.0) | NC |
| HPV 35 | 41 (21.1) | 25 (21.9) | 16 (20.0) | 0.746 |
| HPV 45 | 14 (7.2) | 6 (5.3) | 9 (11.3) | 0.124 |
| HPV 52 | 13 (6.7) | 4 (3.5) | 9 (11.3) | 0.043† |
| HPV 58 | 16 (8.2) | 13 (11.4) | 3 (3.8) | 0.066† |
| HPV 59 | 21 (10.8) | 10 (8.8) | 11 (13.8) | 0.272 |
| HPV 56 | 15 (7.7) | 8 (7.0) | 7 (8.8) | 0.657 |
| Low risk types | | | | |
| HPV 6/11 | 91 (46.9) | 42 (36.8) | 49 (61.3) | 0.001 |
| Vaccine specific types | | | | |
| Bivalent vaccine type HPV 16 & /18 | 107 (55.2) | 75 (65.8) | 32 (40.0) | 0.000 |
| 4-valent vaccine type HPV 6/11/16/18 | 163 (84.0) | 94 (82.4) | 69 (86.2) | 0.478 |
| 9-valent vaccine type‡ HPV 6/11/16/18/31/33/45/52/58 | 181 (93.3)§ | 105 (92.1) | 76 (95.0) | 0.427 |

* $\chi 2$ test was used for p values.
†Fisher's exact test.
‡Includes 9-valent all types which come as alone, in combination with other 9-valent specific types or in combination with non-9-valent HR types such as HPV type 35, when we excluded HPV 35 (n=33).
§93.3% changed to 76.3%.
HPV, human papillomavirus; HR, high-risk; MSM, men who have sex with men; NC, not calculated.

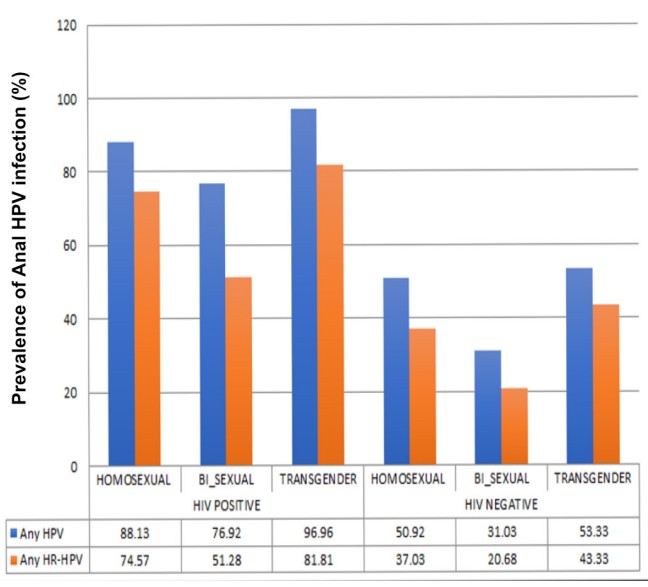

| | Any HPV | Any HR-HPV |
|---|---|---|
| HOMOSEXUAL (HIV POSITIVE) | 88.13 | 74.57 |
| BI_SEXUAL (HIV POSITIVE) | 76.92 | 51.28 |
| TRANSGENDER (HIV POSITIVE) | 96.96 | 81.81 |
| HOMOSEXUAL (HIV NEGATIVE) | 50.92 | 37.03 |
| BI_SEXUAL (HIV NEGATIVE) | 31.03 | 20.68 |
| TRANSGENDER (HIV NEGATIVE) | 53.33 | 43.33 |

**Figure 1** Prevalence of any HPV and HR-HPV by HIV status and sexual orientation. HPV, human papillomavirus; HR, high-risk.

in MSM living with HIV and commented on the fluid nature of sexual identity without mentioning the exact prevalence of MSW among them. Moreover, given the fact that our study has a cross-sectional design, incubation time for infection and variation in viral clearance could be other possible explanations for the differences in the prevalence.[29 38]

For the distribution of anal HPV types, our study results are similar to the studies done elsewhere.[39] HPV16 (35.1%) was the most common anal oncogenic HPV type detected both in MSM living with and without HIV (44.7% vs 21.2%, p<0.001). This is in close approximation of the summary prevalence for HPV16 calculated from a recent meta-analyses[7 40] and by others.[29 33 41 42] HPV16 has the ability to persist in the anal canal more than any other types and not only tends to cause anal intraepethilial neoplasia (AIN)[29] but also associated with disease severity.[7]

The prevalence of HR-HPV type 35 (21.1%) in our study was similar to the levels reported by India (20%),[32] but higher than HPV type 35 levels reported from other parts of the world.[31–33] Given the fact that HPV35 is not

**Table 3** Factors associated with any type of detectable HPV infection and HPV type 16, among MSM and transgender women in Karachi Pakistan (N=298)

| Risk factors | HPV positive N (%) | Any anal HPV infection Crude PR (95% CI) | Any anal HPV infection Adjusted PR (95% CI) | Anal HPV 16—infection Crude PR (95% CI) | Anal HPV 16—infection Adjusted PR (95% CI) |
|---|---|---|---|---|---|
| HIV status | | | | | |
| No (ref) | 80 (48.0) | | 1.0 | 1.0 | 1.0 |
| Yes | 114 (87.0) | 1.81 (1.36 to 2.4) | 1.44 (1.05 to 1.97) | 3.33 (2.55 to 3.51) | 2.81 (2.16 to 3.82) |
| Other STI* | | | | | |
| No (ref) | 61 (31.4) | 1.0 | 1.0 | 1.0 | 1.0 |
| Yes | 133 (68.6.0) | 1.32 (1.08 to 2.13) | 1.29 (1.02 to 1.91) | 1.97 (1.28 to 3.04) | 1.52 (0.99 to 2.37) |
| Preferred anal sex role | | | | | |
| Insertive (ref) | 21 (10.8) | 1.0 | 1.0 | 1.0 | |
| Receptive | 173 (89.2) | 1.95 (1.09 to 3.50) | 1.56 (0.87 to 2.83) | 2.49 (1.02 to 6.14) | |
| Condom use | | | | | |
| Consistent (ref) | 15 (7.7) | 1.0 | 1.0 | 1.0 | 1.0 |
| Inconsistent | 141 (72.7) | 3.30 (1.94 to 5.6) | 2.01 (1.07 to 5.20) | 2.96 (1.48 to 5.92) | 2.61 (1.50 to 4.50) |
| Never | 38 (19.6) | 2.42 (1.04 to 5.21) | 1.58 (1.03 to 4.44) | 3.11 (1.71 to 5.66) | 3.08 (1.69 to 5.60) |
| Smoking | | | | | |
| No | 34 (17.5) | 1.0 | | 1.0 | |
| Yes | 160 (82.5) | 1.34 (0.93 to 1.93) | | 1.17 (0.71 to 1.90) | |

*Trichomonas, gonorrhoea.
†All significant variables were adjusted for age, sociodemographic and all other sexual behaviours.
HPV, human papillomavirus; MSM, men who have sex with men; PR, prevalence ratio; STI, sexually transmitted infection.

included in the FDA-approved 9-valent vaccine, its relatively high prevalence can have implications on anal cancer prevention in our region.

Similar to previous research,[43] our results also revealed that almost 4 out of 5 MSM carried at least one of the HR genotypes in their anal canal (80% MSM living with HIV vs 73% of MSM without HIV infection). This is of concern in HIV-negative MSM in particular, since infection with HR-HPV types is reported to increase the likelihood of acquiring HIV.[16] Thus, our results further emphasise the need to screen Pakistani MSM more frequently for both HPV and HIV given their HR of development of malignant disease.

Our study detected a high prevalence of multiple HPV types—that is, at least 2 types (either LR and HR or HR & HR)—of 47% in HPV positive MSM as a group, which is similar to other, studies.[44] This indeed is a matter of concern as anal infection with multiple HR-HPV types increases the likelihood of an individual to be infected with HPV16 and, infection with HPV16 has been associated with AIN and progression to higher grade AIN over time.[3] Moreover, its prevalence was significantly higher in MSM living with HIV. Higher HPV reinfection rates and lower clearance rates in individuals living with HIV may also play a crucial role in the development and the progression of anal neoplastic disease.[29 45]

The results of our study suggest that the most frequently distributed HPV types among Pakistani MSM could be covered by the currently available nonavalent (93.3%)

and quadrivalent (84%) vaccines, which means that most of HPV-related diseases including cancers and anogenital warts may potentially be preventable if offered to MSM at initiation of sexual debut. However, the 20% carrying HPV35 would not be covered by the currently available vaccines.

More than 50% of HPV-positive MSM were young, below 25 years of age, most were single and sexually very active, and most make a living and support their families through sex work.[46] As many as 80% of HPV-positive MSM had initiated sex work in their teens, which is in line with other studies.[47] The high number of parallel sexual partners may contribute to further spread of HPV and other STIs among MSM.

Among other sexual behaviours, inconsistent or never use of condom was the only variable that was significantly associated with any HPV, specifically HPV16 infection. For many in our study population, negotiating safe sex is not an option as, on an average, a MSW will service 3–5 clients per night with a one-time service charge as low as PKR100–PKR150 (US$0.7–US$1.03),[19 48] though higher rates may be charged for unprotected sex. Thus, financial barriers may impede consistency in condom use. In addition, receptive anal sex was significantly more common among HPV-infected MSM, considering the fact that this is the strong transmission route and poses HR for HIV acquisition.[49] Since the prevalence of anal HPV was more than twice as common in MSM and transgender women living with HIV, this finding indeed turned out

to be consistent with our multiple regression model that identified HIV infection as an independent risk factor for any anal HPV or HPV16 infection confirming MSM and transgender women living with HIV as a priority for the prevention of anal cancer.[40]

## Limitations of the study

Our study had some limitations. First, the design of the study was cross-sectional, yet many HPV infections are non-persistent in nature.[50] Moreover, the cross-sectional design precludes us to determine a temporal relationship, however, the risk factors identified in our study appear robust and have also been found in studies from other countries. Second, our study results may have limited generalisability because participants were recruited from only one city of Pakistan (Karachi) and were recruited using peer referrals and snowball sampling techniques. Nevertheless, Karachi is the largest metropolitan city of Pakistan, representing all ethnicities of the country. Moreover, in a country like Pakistan, which is by and large a heteronormative society and where homophobia prevails against this sexual minority due to cultural and religious inhibitions, using above-mentioned sampling technique was the most feasible option for the recruitment process. Third, we cannot exclude the possibility of under-reporting or over-reporting of sexual behaviours and lifestyle factors that could not only influence the strength of association with the outcome but might also lead to information bias since they were self-reported.

## Recommendations

HIV care and treatment programmes in Pakistan represent a potential window of opportunity for providing preventive clinical interventions to avert an anal cancer epidemic nested within the HIV epidemic. Providing cost-effective, life-saving anal cancer prevention services (ie, screen and treat approach and vaccination) linked to HIV care could not only avert preventable HPV-related malignancies but could also lead to a more efficient utilisation of resources and improve healthcare access for those most at risk. Moreover, our study findings emphasise the need for government support of interventions that can improve access to and quality of services to MSM including MSW. Prevalent cultural and social stigma must also be addressed since this is essential for protecting the dignity and autonomy of this population.

## Conclusions

We believe that our study findings are of importance for clinical practice and public health policy. The relatively high prevalence of HR HPV indicates a future risk of anal cancer among MSM in general and among MSM living with HIV in particular. Given that HPV35 currently is not included in the 9-valent HPV vaccine, we strongly feel that this is one of our most important contributions and that there is a need for regular anal pap screening among MSM and transgender women to enable early detection and treatment to prevent anal cancer.

**Author affiliations**
[1]Department of Global Public Health, Karolinska Institutet, Stockholm, Sweden
[2]Community Health Sciences, Aga Khan University Medical College, Karachi, Sindh, Pakistan
[3]Lab Medicine, Orebro Universitet, Orebro, Sweden
[4]Molecular Biology, Sindh Institute of Urology and Transplantation, Karachi, Sindh, Pakistan
[5]Community Health Sciences & School of Nursing, Aga Khan University Medical College, Karachi, Sindh, Pakistan

**Acknowledgements** We thank our all study participants and the staff at our study sites for their continuous cooperation and collaboration. We are enormously thankful to the Sindh Institute of Urology and Transplantation (SIUT) for their lab supplies (reagents and the sample collection kits) and contribution to the lab analyses. We are grateful to the staff at the histopathology and molecular laboratories of SIUT who devoted their time and efforts to this project (in-kind). We are also grateful for the financial contribution from the Dean's Fund at The Aga Khan Hospital & Medical University which covered the cost of samples transportation to the SIUT labs from the two study sites. Special thanks to Dr Asli Kulane for her encouragement and guidance to conduct this study. Thanks to Ms Shazia for her assistance in the conduct of the lab work. Special thanks to our English Editor Ms Carla Sturm for reviewing the manuscript for typographical/grammatical errors.

**Contributors** ME and SB conceptualised the study. ME designed and conducted the study. AME, SA and TA supervised the main study implementation. Lab work was supervised by SB and ME. ME did the data collection, data cleaning and mining. ME did the data analysis. ME prepared the first draft of the manuscript with initial revision from SA and TA. AME did the extensive revisions of the manuscript with ME. All authors reviewed and approved the final version of the manuscript. ME, as guarantor takes the full responsibility for the work and the conduct of the study.

**Funding** Global and Sexual Health Research Group Department of Global Public Health Karolinska Institutet Stockholm Sweden (Org number:202100-2973; Vat number: SE202100297301).

**Competing interests** None declared.

**Patient and public involvement** Patients and/or the public were not involved in the design, or conduct, or reporting, or dissemination plans of this research.

**Patient consent for publication** Not applicable.

**Ethics approval** The Ethics Committee on Human Research of Aga Khan (3612-CHS-ERC-15) and Dow (IRB-557/DUHS/APPROVAL/2015/84) Universities of Health Sciences approved the study.

**Provenance and peer review** Not commissioned; externally peer reviewed.

**Data availability statement** Data are available on reasonable request. The data that support the findings of this study are available from Global & Sexual Health Research Group of Department of Global Public Health on reasonable request and with permission of Karolinska Institutet (KI) Stockholm Sweden.

**ORCID iD**
Muslima Ejaz http://orcid.org/0000-0002-1187-9095

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
