## [Reviewer comments · BMJ Open]

ARTICLE DETAILS

TITLE (PROVISIONAL)	Anal human papillomavirus infection among men who have sex with men and transgender women living with and without HIV in Pakistan: Findings from a cross-sectional study
AUTHORS	Ejaz, Muslima; Andersson, Soren; Batool, Salma; Ali, Tazeen; Ekström, Anna

VERSION 1 – REVIEW

REVIEWER	Alizon, Samuel Centre National de la Recherche Scientifique (CNRS), MIVEGEC
REVIEW RETURNED	05-May-2021

GENERAL COMMENTS	Summary The authors develop a cross-sectional study to analyse the prevalence of human papillomavirus (HPV) anal infections in men having sex with men (MSM) in Pakistan. Consistently with studies conducted in other countries, they find significant effects of age, smoking status, number of partners, condom use, and other STIs on HPV carriage. Still consistent with existing studies, the most prevalent HPV genotype is HPV16. Finally, they find marked differences between HIV-positive and HIV-negative MSMs. These results have implications for the management of HPV infections in Pakistan in the context of vaccination. Review This data represents a valuable addition to the literature. However, I do have some concerns about the writing and the statistical analyses. 1) Statistical analyses Overall, I was a bit lost with the statistical analyses performed by the authors. On page 6 line 25 for instance, it is unclear how they perform a confidence interval on a proportion. On the same page in line 50, it is unclear what the p-value refers to. More generally, the authors are performing so many different tests that a Bonferroni correction might be necessary. In my opinion, it would be simpler if the authors built a unique multivariate generalised linear model assessing the risk to be infected or not by HPV. They would then be able to reproduce Table 1 with odds ratios associated with the different factors included. Furthermore, I think Table 4 could then be merged with
--

	Table 1 (unless the authors specifically want to study interactions between factors). Another Supplementary Table could then be performed for HPV16 positivity more specifically. One point that I found unclear is where did the enrolment take place. Were there multiple clinics? If so, this should probably be included as a factor in the analysis for Table 1. Furthermore, adding the sampling date (after centring and reducing the values) in the analysis might also be a good idea. Finally, about Tables 2 and 3, it would be good to indicate whether the distributions are significant or not, especially with regards to sexual orientation in Table 3. 2) Reference to prior work In general, I think the reference to prior work could be improved. First, the quality of the bibliography is clearly below average (many references even lack journal names). Second, the authors often seem to be citing peripheric studies when more thorough analyses exist (see for instance the work from the WHO International Agency for Research on Cancer, IARC). Third, some reference could be updated, e.g. reference number 1 from 2005 about HPV being the most prevalent STI. 3) Potential limitations The authors mention that several infections are "transient". It would be good to be more specific here and explain whether this refers to non-persisting infections (e.g. Alizon et al 2017 Viruses), latent infections (e.g. Gravitt & Winer 2017 Viruses), or carriage with detection of the HPV from the partner. 4) Writing and Figures I found that in some places the writing could be improved. There are also some typos. I would advise the authors to refrain from using formulations such as "the first ever" because it is difficult to prove and it brings little addition to the quality of the study. I think the 3D figures are not necessary here and using simpler plots will be easier to read.
--	---

REVIEWER	Clifford, Gary International Agency for Research on Cancer
REVIEW RETURNED	13-Jul-2021

GENERAL COMMENTS	The authors should be congratulated for their work to obtain this unique dataset on HPV prevalence in such a high risk, and hard to reach, population. This is the first such report on MSM in this part of the world, and is a sizeable and informative sample. Study design is appropriate and relevant. It is of considerable public health importance for anal cancer prevention. That being said, I do have some major suggestions to improve data presentation and interpretation to make these data of more utility for the wider scientific community:
--

1) Abstract: The objectives could be shortened to allow more emphasis on results. In particular, I would have liked to see the effect of HIV status (and other significant factors , e.g. condom use) on anal HPV16, by far the most relevant type for anal cancer prevention.

abstract conclusion: The sentence "Current findings support HPV vaccination efforts to this high-risk population." is unfortunately not true. Such a high HPV prevalence means it is too late for prophylactic vaccination (which needs to occur before sexual activity) in this population. Rather, these data show that only a gender neutral approach prior to sexual debut can help prevent anal cancer in future generations of MSM. It is a disservice to the MSM community to simply insist on catch up vaccination as a useful tool - almost all anal cancers in this population will arise in men that have prevalent HPV infection.

HPV35 : In various parts of the manuscript (including strengths and conclusions), it is stated that HPV35 is important for vaccination and anal cancer prevention in this high-risk population. Yet this study is about HPV infection, not cancer, so this is an over-interpretation. Non-HPV16 types can be very common, but only rarely cause cancer, even in HIV-positive persons (Lin et al, Lancet Infect Dis, 2017). HPV vaccines are ultimately designed to prevent cancer.

Introduction :is generally a bit long, and could be made more concise, for example by reducing issues that are also addressed in the discussion, and by removing details (e.g. number of geographical spots where MSM were mapped to in Pakistan).

At line 13, suggest to remove particular reference HPV18 if talking also about anus and oropharynx. HPV18 does not stand out beyond other HR hPV types for the anus (Lin et al, 2017).

Statistical analysis: The four sentences on a priori sample size calculations are not appropriate for the reporting of study end results. Rather rely upon statistical tests and confidence intervals.

Results : HPV DNA Prevalence and genotype distribution: There is a missing comparator prevalence of any oncogenic anal HPV infection at line 28.

Sexual orientation of the study sample should be described before results of HPV prevalence.

Table 1 – It would be more appropriate and reader friendly to show HPV prevalence (N/n; row %) for each category of characteristics (as is the case for the category "other STI"). Means (age etc) can be reported in the text only.

Table 2 – Major Column headers would be more reader friendly as simply HIV positive (N=131), HIV-negative (N=167) and all (N=298). The percentages are unclear –suggest that all percentages in the table and throughout the text should be shown with all men as denominator (not HPV positive men only).

Table 3 – too much precision in the percentages (i.e. too many decimal points), and no statistical comparisons. As per my comment for Table 1, I suggest it would be of more interest to see HPV prevalence by sexuality and HIV status, with statistical tests for differences in HV prevalence. If no difference, then perhaps

	numbers are too small, or of not enough interest to merit a full table and can be mentioned in text only. Table 4. Rather than showing column percentages in the Overall column, more intuitive to show N/n HPV pos (%HPV pos). It is also unclear why there is only a p-value for crude and not adjusted PR. And only for HR-HPV not HPV16. Either drop completely (and rely on PR 95% CIs) or be consistent throughout the table. Figure 1. Redundant with Table 2. No need for both. Figure 2. Redundant with Table 3. No need for both. Discussion – Rather than over-analyse similarities/differences with a group of cherry picked studies of HPV prevalence, better to compare with relevant meta-analysis data (e.g. Marra et al JID, 2019). The discussion about “MSWs who accounted for 80% of our study population” is unclear. I think I understand what the authors are trying to say about definitions of sexual preference being arbitrary (especially in the scenario of societal repression), so best to avoid using such strong label/definitions themselves. “Given the fact that HPV 35 is not included in the FDA-approved 9-valent vaccine its relatively high prevalence can have implications on anal cancer prevention in our region.” – see comment above about why this is over-interpretation “This is of concern in HIV-negative MSM in particular, since infection with HR HPV type is reported to increase the likelihood of acquiring HIV37,38. Thus, our results indicate a high risk of development of malignant disease related to HPV infection among MSM in Pakistan.” 1) the cited data are from the cervix in women. 2) HIV infection is not a malignant disease (maybe there is not supposed to be a direct link to the previous sentence ? but unclear). “This indeed is a matter of concern as anal infection with multiple HPV types has been associated with AIN and progression to higher grade AIN over time3.” This is not really true, either in the cervix or the anus. The greater the number of types, the more chance of having HPV16 – but it is not “multiplicity” per se that is more dangerous.
--	--

VERSION 1 – AUTHOR RESPONSE

Reviewer:

1

General comment: The authors develop a cross-sectional study to analyse the prevalence of human papillomavirus (HPV) anal infections in men having sex with men (MSM) in Pakistan. Consistently with studies conducted in other countries, they find significant effects of age, smoking status, number of partners, condom use, and other STIs on HPV carriage. Still consistent with existing studies, the most prevalent HPV genotype is HPV16. Finally, they find marked differences between HIV-positive and HIV-negative MSMs. These results have implications for the management of HPV infections in Pakistan in the context of vaccination.

Response: Thank you very much for this encouraging comment.

Review:

1) Statistical analyses

Comment a) Overall, I was a bit lost with the statistical analyses performed by the authors. On page 6 line 25 for instance, it is unclear how they perform a confidence interval on a proportion.

Response: Thank you for this comment. We calculated the confidence interval using the following formula for confidence intervals (CI) = $p \pm z \cdot \sqrt{p(1-p) / n}$, and I think it adds information to the reader on the precision of our estimates.

It is quite common to estimate a CI interval around a prevalence, for example made by Lin et al who reported on HPV prevalence rates in a systematic review¹ and also by Marra et al².

Comment b) On the same page in line 50, it is unclear what the p-value refers to. More generally, the authors are performing so many different tests that a Bonferroni correction might be necessary.

The p-value indicates that there are statistically significant differences between subgroups of participants stratified on HIV status on our study

We are thankful to the reviewer who has suggested us for the Bonferroni correction method. We applied to look at any statistical difference on the prevalence of "any HPV" and "any HR-HPV" stratified on HIV status and sexual orientation (homosexuals, bisexuals and transgender women). The results section is further explaining that the homosexuals living with and without HIV are statistically different from bisexuals and transgender women for the prevalence of "any HPV and "any HR HPV"

This information has also been clarified in the Results section.

¹ "Human papillomavirus types from infection to cancer in the anus"

<https://www.semanticscholar.org/paper/Human-papillomavirus-types-from-infection-to-cancer-Lin-Franceschi/0421ca71dcfbdde03c9889f840305295b1a5aa3c>. Accessed 9 Aug. 2021.

² "Type-Specific Anal Human Papillomavirus Prevalence ... - PubMed." 29 Jan. 2019,

<https://pubmed.ncbi.nlm.nih.gov/30239749/>. Accessed 12 Aug. 2021.

Comment c) In my opinion, it would be simpler if the authors built a unique multivariate generalized linear model assessing the risk to be infected or not by HPV. They would then be able to reproduce Table 1 with odds ratios associated with the different factors included.

Response: Since this is a cross-sectional study with a binary outcome, we applied a Cox proportional hazard model algorithm using robust SE, which we believe is the most appropriate model building technique and estimated the association between the exposure and the outcome using means of prevalence ratios (PRs).

We agree that when adjustments for potential confounders are needed, logistic regression models are commonly used to estimate odds ratios (ORs) that are reported in a similar way as PR. However, OR are less suitable when the outcome is very common like in our study. In these situations, interpreting ORs as if they were PRs may be inadequate.³

(ref: Barros AJ, Hirakata VN. Alternatives for logistic regression in cross-sectional studies: an empirical comparison of models that directly estimate the prevalence ratio. BMC Med Res Methodol. 2003;3:21. DOI: 10.1186/1471-2288-3-21)

For example, if 80 out of 100 exposed subjects have a particular disease and 50 out of 100 non-exposed subjects have the disease, then the odds ratio (OR) is $(80/20)/(50/50)=4$. However, the prevalence ratio (PR) is $(80/100)/(50/100)=1.6$. The latter indicates that the exposed subjects are only 1.6 times as likely to have the disease as the non-exposed subjects, and this is the number in which most people would be interested.

There is considerable literature on the advantages and disadvantages of OR versus PR that we

³ "Alternatives for logistic regression in cross-sectional studies: an" 20 Oct. 2003, <https://pubmed.ncbi.nlm.nih.gov/14567763/>. Accessed 9 Aug. 2021.

believe supports our choice of using the PR. (Greenland, Stromberg, Axelson et al and others)^{4,5,6,7,8}.

1. Greenland S. *Interpretation and choice of effect measures in epidemiologic studies. Am J Epidemiol* 1987;125:761–8.

2. Stromberg U. *Prevalence odds ratio v prevalence ratio. Occup Environ Med* 1994;51:143–4.

3. Axelson O, Fredricksson M, Ekberg K. *Use of the prevalence ratio v the prevalence odds ratio as a measure of risk in cross sectional studies. Occup Environ Med* 1994;51:574.

Thus, given that the outcome of interest (HPV infection) was so common (i.e., outcome >10%), we prefer to use PRs and hope that the reviewer agrees with us. The reason for using PR, has now been explained a bit better in the Methods section under the heading of data management and data analysis. Page, 7, Line 186-191.

Comment d) Furthermore, I think Table 4 could then be merged with Table 1 (unless the authors specifically want to study interactions between factors).

Response: Here we respectfully would prefer to keep the tables separately, also in order to adjust to the comments from reviewer 2 who rather suggests highlighting our findings on HPV 16 as an outcome variable in the abstract, reflecting the results.

Comment. Another Supplementary Table could then be performed for HPV16 positivity more specifically.

Response: See above.

⁴ "Methods for estimating prevalence ratios in cross- sectional studies." 27 Nov. 2007, https://www.scielo.br/pdf/rsp/v42n6/en_7118.pdf. Accessed 9 Aug. 2021.

⁵ "Interpretation and choice of effect measures in epidemiologic analyses." <https://pubmed.ncbi.nlm.nih.gov/3551588/>. Accessed 9 Aug. 2021.

⁶ "Prevalence odds ratio v prevalence ratio - PubMed." <https://pubmed.ncbi.nlm.nih.gov/7710464/>. Accessed 9 Aug. 2021.

⁷ "Use of the prevalence ratio v the prevalence odds ratio ... - NCBI - NIH." <https://www.ncbi.nlm.nih.gov/pmc/articles/PMC1128127/>. Accessed 9 Aug. 2021.

⁸ "Use of the prevalence ratio v the prevalence odds ratio as ... - PubMed." <https://pubmed.ncbi.nlm.nih.gov/7951785/>. Accessed 9 Aug. 2021.

Comment. One point that I found unclear is where did the enrollment take place. Were there multiple clinics? If so, this should probably be included as a factor in the analysis for Table 1.

Response: Thank you for pointing out the need to clarify this better. In fact we experienced quite a challenging situation during our data collection period at a public hospital in terms of space, privacy and respect for the integrity of our study participants. The data collection procedure included not only filling the questionnaire but also collection of anal swabs. For anal swab collection we were given two days a week only and we had to rely on the space and staff of the overcrowded and overwhelmed public hospital in terms of the service provision. Additionally, as a result of heteronormative mindset of our society, our participants were receiving inappropriate and unpleasant remarks from the staff. It is therefore we had to relocate our data collection setting to CBO working for this sexual and gender minority. We trained STI doctor from their sexual health clinic for the collection of anal swabs. We have also clarified this even further in the Methods section to avoid misunderstandings. We have also followed your suggestion and included information of where the participants were recruited in Table 1.

Furthermore, adding the sampling date (after centering and reducing the values) in the analysis might also be a good idea.

Response: To reach this very hard to reach target population, we had to use snow-ball sampling to collect cross-sectional data. The initial recruitment started at the ART clinic with transgender women. Thereafter, with a few months delay, we expanded to include MSM from a CBO. Thus, any influence from sampling date has been taken care by adjusting for sexual orientation. Moreover, the study participants enrolled during two phases of the data collection were compared on different sociodemographic and sexual and behavioral characteristics. No significant difference was observed and it is therefore, they were merged for analysis.

This has now been better explained in the Methods and the Discussion section.

Comment: Finally, about Tables 2 and 3, it would be good to indicate whether the distributions are significant or not, especially with regards to sexual orientation in Table 3.

Response: Thank you for this comment, we have added the p values and the result of these comparisons are now discussed in the Results section on page 9.

2) Reference to prior work

In general, I think the reference to prior work could be improved. First, the quality of the bibliography is clearly below average (many references even lack journal names). Second, the authors often seem to be citing peripheric studies when more thorough analyses exist (see for instance the work from the

WHO International Agency for Research on Cancer, IARC). Third, some reference could be updated, e.g. reference number 1 from 2005 about HPV being the most prevalent STI.

Response: Thanks for noticing this mistake. We have improved the quality of the bibliography list and added the names of the journals. We have also updated the reference list to include the latest and most relevant citations, replacing the older citations ones.

3) Potential limitations

Comment: The authors mention that several infections are "transient". It would be good to be more specific here and explain whether this refers to non-persisting infections (e.g. Alizon et al 2017 Viruses), latent infections (e.g. Gravitt & Winer 2017 Viruses), or carriage with detection of the HPV from the partner.

Response: Yes we agree with the reviewer. By "transient", we did mean "non-persistent" as described by Alizon 2017, so we corrected this sentence and cited Alizon et al.

4) Writing and Figures

Comment: I found that in some places the writing could be improved. There are also some typos. I would advise the authors to refrain from using formulations such as "the first ever" because it is difficult to prove, and it brings little addition to the quality of the study.

Response: thank you for noticing these mistakes. We think we have corrected them all, and an English editor has reviewed the revised manuscript.

Comment: I think the 3D figures are not necessary here and using simpler plots will be easier to read.

Response: We have replaced 3D with simple plots

Reviewer: 2

Comments to the Author:

Comment: The authors should be congratulated for their work to obtain this unique dataset on HPV prevalence in such a high risk, and hard to reach, population. This is the first such report on MSM in this part of the world, and is a sizeable and informative sample. Study design is appropriate and relevant. It is of considerable public health importance for anal cancer prevention.

Response: Thank you for this encouraging comment.

That being said, I do have some major suggestions to improve data presentation and interpretation to make these data of more utility for the wider scientific community:

- 1) Abstract: The objectives could be shortened to allow more emphasis on results. In particular, I would have liked to see the effect of HIV status (and other significant factors, e.g. condom use) on anal HPV16, by far the most relevant type for anal cancer prevention.

Response: Thank you for this suggestion. We agree that the fact that we have analyzed HPV 16 status is a strength of this study and of great importance. Thus we have added information on this in the abstract.

Abstract conclusion: The sentence "Current findings support HPV vaccination efforts to this high-risk population." is unfortunately not true. Such a high HPV prevalence means it is too late for prophylactic vaccination (which needs to occur before sexual activity) in this population. Rather, these data show that only a gender-neutral approach prior to sexual debut can help prevent anal cancer in future generations of MSM. It is a disservice to the MSM community to simply insist on catch up vaccination as a useful tool - almost all anal cancers in this population will arise in men that have prevalent HPV infection.

Response: We entirely agree with the reviewer on this issue. Thank you for highlighting this issue. We have rephrased our conclusion to:

The results of our study may be of public health importance and indicate the need for a gender-neutral HPV vaccination approach. Reaching out to all young people with HPV vaccination, before sexual debut, would help prevent anal cancer not only in women but also in future generations of MSM and transgender women.

- HPV 35: In various parts of the manuscript (including strengths and conclusions), it is stated that HPV35 is important for vaccination and anal cancer prevention in this high-risk population. Yet this study is about HPV infection, not cancer, so this is an over-interpretation. Non-HPV16 types can be very common, but only rarely cause cancer, even in HIV-positive persons (Lin et al, Lancet Infect Dis, 2017). HPV vaccines are ultimately designed to prevent cancer.

Response: We completely agree with the reviewer that so far HPV 16 is the most common genotype that has shown its causal association with anal cancer, however, the subtypes, having oncogenic potential are HPV subtypes 16, 18, 31, 33, 35, 39, 45, 51, 52, 56, 58, 66 and 69

Unfortunately HPV 35 is under-researched area and to date only few studies have found this genotype in MSM both living with and without HIV, however, since this subtype has oncogenic potential one can assume that it could be a potential driver towards development of precancerous lesions as supported by some recent research e.g. (Eunhyang Park (2019) A Korean study⁹ on carcinogenic risk of HPV genotypes where he has reported HPV 35 has 2.65 times risk for HSIL)

A study from India among MSM also reported that HPV 35 may be of oncogenic importance¹⁰ and concluded as: (*Vaccine based prevention strategies for HPV infection in India should consider potential differences in HPV type distribution among HIV-infected MSM when designing interventions.*)

Given that HPV 35 currently is not included in the 9-valent HPV vaccine, we strongly feel that this is one of our most important contributions and that there is a need for regular anal pap screening among MSM and transgender women to enable early detection and treatment to prevent anal cancer. We have emphasized this a bit more in Discussion/Conclusions.

- 2) Introduction: is generally a bit long, and could be made more concise, for example by reducing issues that are also addressed in the discussion, and by removing details (e.g. number of geographical spots where MSM were mapped to in Pakistan).

Response: We agree with the reviewer and have edited the introduction to make it shorter.

- 3) At line 13, suggest removing particular reference HPV18 if talking also about anus and oropharynx. HPV18 does not stand out beyond other HR HPV types for the anus (Lin et al, 2017).

Response: We agree, and we have removed and rephrased those wordings in the beginning of the introduction.

- 4) Statistical analysis: The four sentences on a priori sample size calculations are not appropriate for the reporting of study end results. Rather rely upon statistical tests and confidence intervals.

Response: We agree with the reviewer and have given separate heading for sample size calculation. Thereafter comes the heading for data management and data analysis where we have given a more detailed description of what statistical test we have used.

⁹ "Carcinogenic risk of human papillomavirus (HPV) genotypes and" 29 Aug. 2019, <https://www.nature.com/articles/s41598-019-49060-w>. Accessed 9 Aug. 2021.

¹⁰ "Prevalence of anal HPV infection among HIV-positive ... - NCBI - NIH." <https://www.ncbi.nlm.nih.gov/pmc/articles/PMC4939069/>. Accessed 9 Aug. 2021.

Results: HPV DNA Prevalence and genotype distribution: There is a missing comparator prevalence of any oncogenic anal HPV infection at line 28.

Response: Thank you for pointing out this mistake, we have added the p-value

Sexual orientation of the study sample should be described before results of HPV prevalence.

Response: Thank you. We have moved the paragraph on sexual orientation, it now appears before the paragraph on HPV DNA prevalence.

- 5) Table 1 – It would be more appropriate and reader friendly to show HPV prevalence (N/n; row %) for each category of characteristics (as is the case for the category “other STI”). Means (age etc) can be reported in the text only.

Response: Thank you for this excellent suggestion. We have added details of the quantitative variables in the Results section and edited table 1 according to the reviewer’s suggestions.

- 6) Table 2 – Major Column headers would be more reader friendly as simply HIV positive (N=131), HIV-negative (N=167) and all (N=298). The percentages are unclear –suggest that all percentages in the table and throughout the text should be shown with all men as denominator (not HPV positive men only).

Response: Thank you for this suggestion, we agree with the comment from the reviewer and have tried to make the text more reader-friendly by saying HPV among HIV infected and HIV uninfected.

However, for the proportions of type specific HPV calculations we have made calculations based on the conditional probability rule (similar to Lin et al (2017)). The details of this have now been described in the Methods section under “Statistical analysis” and reads as follow;

"The overall HPV DNA prevalence (as determined by PCR) is reported as the proportion of HPV positive participants among all included individuals (In our case 194/298). Conversely, type-specific HPV positivity is presented as the proportion of HPV-subtype positive individuals divided by all HPV-positive individuals (in our case X/194).

- 7) Table 3 – too much precision in the percentages (i.e. too many decimal points), and no statistical comparisons. As per my comment for Table 1, I suggest it would be of more interest to see HPV prevalence by sexuality and HIV status, with statistical tests for differences in HPV prevalence. If no difference, then perhaps numbers are too small, or of not enough interest to

merit a full table and can be mentioned in text only.

Response: Thank you for pointing this out. Table 2 provides information on HPV overall and on all other tested genotypes stratified by HIV status. We have edited the table according to the suggestion of the reviewer, and have added a column for p-value (Table 2).

However, for Table 3, the sample size was too limited to list HPV genotype prevalence by sexual orientation also stratifying by HIV status. We tried this but the numbers became too small to draw any valid conclusions (resulting in empty cells empty or fewer than 5 observations per cell).

Thus, we grouped the data and analyzed “any oncogenic HPV type” instead, by sexual orientation, as described in original table 3 and figure 2. Given the small numbers, we suggest following the recommendation of Reviewer 2, and only include figure 2 instead of both Figure 2 and Table 3.

- 8) Table 4. Rather than showing column percentages in the Overall column, more intuitive to show N/n HPV pos (%HPV pos). It is also unclear why there is only a p-value for crude and not adjusted PR. And only for HR-HPV not HPV16. Either drop completely (and rely on PR 95% CIs) or be consistent throughout the table.

Response: Thank you for this suggestion, we have followed the suggestion and have added percentages for the HPV positive. We have also deleted the p-value column for HR-HPV to be consistent.

- 9) Figure 1. Redundant with Table 2. No need for both.
Figure 2. Redundant with Table 3. No need for both.

Response: we have followed the suggestion from the reviewer and deleted Figure 1 and kept table 2, deleted Table 3, but kept Figure 2 and now it is figure 1.

- 10) Discussion – Rather than over-analyse similarities/differences with a group of cherry-picked studies of HPV prevalence, better to compare with relevant meta-analysis data (e.g. Marra et al JID, 2019).

Response: We have followed the reviewer’s comment and added more recent citations as suggested.

12) The discussion about “MSWs who accounted for 80% of our study population” is unclear. I think I understand what the authors are trying to say about definitions of sexual preference being arbitrary (especially in the scenario of societal repression), so best to avoid using such strong label/definitions themselves.

Response: Thank you for pointing out the need to clarify this a bit more. During the data collection procedure, carried out through face-to-face interviews, after building trust between the interviewer and the respondent, the respondents themselves clearly informed us about what sexual orientation they choose to identify with.

“Given the fact that HPV 35 is not included in the FDA-approved 9-valent vaccine its relatively high prevalence can have implications on anal cancer prevention in our region.” – see comment above about why this is over-interpretation

Response: We are grateful for this thorough review (see previous response above) and we have also modified the text in the conclusion of the abstract

“This is of concern in HIV-negative MSM in particular, since infection with HR HPV type is reported to increase the likelihood of acquiring HIV (37,38). Thus, our results indicate a high risk of development of malignant disease related to HPV infection among MSM in Pakistan.”

1) The cited data are from the cervix in women.

2) HIV infection is not a malignant disease (maybe there is not supposed to be a direct link to the previous sentence? but unclear).

3) *“This indeed is a matter of concern as anal infection with multiple HPV types has been associated with AIN and progression to higher grade AIN over time ³.”*

This is not really true, either in the cervix or the anus. The greater the number of types, the more chance of having HPV16 – but it is not “multiplicity” per se that is more dangerous.

Responses:

1) We have changed the citation to make this clearer.

2) The paragraph starts like that:

“Similar to previous research ³⁶ our results also revealed that almost 4 out of 5 MSM carried at least one of the high-risk genotypes in their anal canal (80% among MSM living with HIV versus 73% among MSM without HIV infection). This is of concern for HIV-negative MSM in particular, since infection with a HR HPV type, may increase the likelihood of acquiring HIV (references have been changed too). Thus, our results further emphasize the need to screen Pakistani MSM more frequently for both HPV and HIV given their high risk of development of malignant disease.”

3) Yes, we agree and have changed the text accordingly.

VERSION 2 – REVIEW

REVIEWER	Alizon, Samuel Centre National de la Recherche Scientifique (CNRS), MIVEGEC
REVIEW RETURNED	01-Sep-2021

GENERAL COMMENTS	I think the manuscript was much improved during the reviewing process and I thank the authors for their efforts in taking into consideration all of my remarks. I only have few comments. page 7 (numbers refer to the highlighted version): I am still puzzled as to why accounting for sexual orientation corrects for the sampling facility. Also, is it possible to discriminate between the two factors? page 10-11: when mentioning a p-value, please always mention the test used. page 11: when calculating a confidence interval on binary factors (e.g. sex or positivity), please indicate in the Methods that you use a binomial proportion confidence interval assuming a Gaussian error (I'm still not sure how relevant it is for sex though). page 20 (and elsewhere): this is a detail but I think the official guidelines are not to put a space between "HPV" and the genotype number, i.e. write HPV16 and not HPV 16. page 33: Why hasn't the study been submitted to a central database such as clinicaltrials.gov? page 35: in terms of research reproducibility, I find that more of the data used and the scripts could be published along with the research because the stratification is such that there does not appear to be risks in terms of patient anonymity.
---

REVIEWER	Clifford, Gary International Agency for Research on Cancer
REVIEW RETURNED	23-Aug-2021

GENERAL COMMENTS	The authors have made a serious and successful attempt to address the concerns of the reviewers. The presentation of the paper is vastly improved and merits publication.
---

	I only have a few remaining small comments: Abstract: "HIV-status (PR: 1.81 95% CI 1.16-2.82), and never-condom-use (PR:2.31, 95% CI 1.03-5.20) were independently associated with prevalence of any-type anal-HPV-infection anal HPV 16 infection" - I believe that these PRs still refer to any HR HPV not HPV16. Need to change. Methods: I repeat my suggestion that the sample size calculation is unnecessary and suggest to completely delete.
--	--

VERSION 2 – AUTHOR RESPONSE

Reviewer:

1

Comments to the Author:

I think the manuscript was much improved during the reviewing process and I thank the authors for their efforts in taking into consideration all my remarks. I only have few comments.

Response: Thank you very much for this comment.

page 7 (numbers refer to the highlighted version): I am still puzzled as to why accounting for sexual orientation corrects for the sampling facility. Also, is it possible to discriminate between the two factors?

Response: We do agree with you, however as we have mentioned in the recruitment section under the heading of Methods that data collection occurred in two phases and had a **time lag** of almost one year. It was only possible to recruit 33 transgender women who were coming to the ART center either for the collection of their ART or for testing and counselling. The rest of the sample of transgender women was recruited from community-based organization after almost one year.

Page 10-11: when mentioning a p-value, please always mention the test used.

Thank you for this suggestion. We, have added the name of the test before each p-value and mentioned the type of tests in the statistical analysis section

Page 11: when calculating a confidence interval on binary factors (e.g. sex or positivity), please indicate in the Methods that you use a binomial proportion confidence interval assuming a Gaussian error (I'm still not sure how relevant it is for sex though).

Response: thank you for this suggestion. We have now clarified in the Results section.

“The overall HPV DNA prevalence (as determined by PCR) is reported as the proportion of HPV positive participants among all included individuals. Conversely, type-specific HPV positivity is presented as the proportion of HPV- subtype positive individuals divided by all HPV positive individuals⁷. The confidence intervals for the proportion were calculated using binomial distribution assuming a Gaussian error

We have calculated confidence interval for just positivity of HPV DNA in the study population.

Page 20 (and elsewhere): this is a detail but I think the official guidelines are not to put a space between "HPV" and the genotype number, i.e. write HPV16 and not HPV 16.

Response: Thank you for pointing this out! Corrected throughout.

Page 33: Why hasn't the study been submitted to a central database such as clinicaltrials.gov?

Response: yes, we could have done this.

page 35: in terms of research reproducibility, I find that more of the data used and the scripts could be published along with the research because the stratification is such that there does not appear to be risks in terms of patient anonymity.

Response: Yes, that can be done.

Comments to the Author:

The authors have made a serious and successful attempt to address the concerns of the reviewers. The presentation of the paper is vastly improved and merits publication

Response: Thank you for this encouraging comment. Indeed, grateful to both of my reviewers that because of their valuable comments our manuscript has improved a lot!

I only have a few remaining small comments:

Abstract: "HIV-status (PR: 1.81 95% CI 1.16-2.82), and never-condom-use (PR:2.31, 95% CI 1.03-5.20) were independently associated with prevalence of any-type anal-HPV-infection anal HPV 16 infection" - I believe that these PRs still refer to any HR HPV not HPV16. Need to change.

Response: Indeed, much grateful for pointing out this mistake. We have now changed the values for "HPV16 infection"

Methods: I repeat my suggestion that the sample size calculation is unnecessary and suggest to completely delete.

Response: We respect your suggestion, and if not mentioning the sample size calculation in the manuscript is acceptable to the journals' guidelines, we are more than willing to delete that portion.

VERSION 3 – REVIEW

REVIEWER	Alizon, Samuel Centre National de la Recherche Scientifique (CNRS), MIVEGEC
REVIEW RETURNED	01-Oct-2021

GENERAL COMMENTS	I think the authors for addressing my final concerns. However, I apologize in advance for this, but I find the first paragraph of the Methods unclear (lines 121-129). I will copy it here for clarity: Enrollment occurred in two phases for this cross-sectional study. The initial recruitment (from March 2016 to May 2016) started at the ART clinic with transgender women, a center run by the National AIDS Control Program of Pakistan in Civil Hospital Karachi (CHK) - the largest (1,900 bed) tertiary care public hospital of
---

	Karachi Pakistan (recruited n=33). Low pace, logistic issues and privacy concerns led to the relocation and setting-up of recruitment center at a sexual health clinic run by a community-based organization, Perwaaz Trust Karachi Pakistan, where the study population was reached using snowball technique and peer referrals. During the second phase between April 2017 to November 2017 a total of 265 MSM and transgender women living with and without HIV were recruited. Thus, any influence from sampling date has been taken care of by adjusting for sexual orientation. As indicated in my previous reviews, the last sentence is a puzzle to me. First, there seems to be an identifiability issue between the date and the sampling facility. Second, I do not understand how sexual orientation corrects for this since the first facility recruited transgender women and the second facility recruited both MSM and transgender women. As an aside, does the "second phase" correspond to the recruitment in the second facility? Finally, I think the English could benefit from some proof-reading before publication.
--	---

VERSION 3 – AUTHOR RESPONSE

Reviewer: 1

Dr. Samuel Alizon, Centre National de la Recherche Scientifique (CNRS)

Comments to the Author:

I think the authors for addressing my final concerns.

However, I apologize in advance for this, but I find the first paragraph of the Methods unclear (lines 121-129). I will copy it here for clarity:

Enrollment occurred in two phases for this cross-sectional study. The initial recruitment (from March 2016 to May 2016) started at the ART clinic with transgender women, a center run by the National AIDS Control Program of Pakistan in Civil Hospital Karachi (CHK) - the largest (1,900 bed) tertiary care public hospital of Karachi Pakistan (recruited n=33). Low pace, logistic issues and privacy concerns led to the relocation and setting-up of recruitment center at a sexual health clinic run by a community-based organization, Perwaaz Trust Karachi Pakistan, where the study population was reached using snowball technique and peer referrals. During the second phase between April 2017 to November 2017 a total of 265 MSM and transgender women living with and without HIV were recruited. Thus, any influence from sampling date has been taken care of by adjusting for sexual orientation.

As indicated in my previous reviews, the last sentence is a puzzle to me. First, there seems to be an identifiability issue between the date and the sampling facility. Second, I do not understand how sexual orientation corrects for this since the first facility recruited transgender women and the second facility recruited both MSM and transgender women.

As an aside, does the "second phase" correspond to the recruitment in the second facility?

Finally, I think the English could benefit from some proof-reading before publication.

Reviewer: 1

Competing interests of Reviewer: None

Response: We have realized that the last sentence is causing confusion. So, we think that the last sentence should be deleted from the manuscript, moreover, further information is added (line 128-130).

Yes, Reviewer is right "second phase" correspond to the recruitment in the second facility

Our manuscript is once again being reviewed by an English Editor

Muslima Ejaz

Corresponding author